# Exploring the Performance of Ultrasound Risk Stratification Systems in Thyroid Nodules of Pediatric Patients

**DOI:** 10.3390/cancers13215304

**Published:** 2021-10-22

**Authors:** Lorenzo Scappaticcio, Maria Ida Maiorino, Sergio Iorio, Giovanni Docimo, Miriam Longo, Anna Grandone, Caterina Luongo, Immacolata Cozzolino, Arnoldo Piccardo, Pierpaolo Trimboli, Emanuele Miraglia Del Giudice, Katherine Esposito, Giuseppe Bellastella

**Affiliations:** 1Division of Endocrinology and Metabolic Diseases, University of Campania “L. Vanvitelli”, 80138 Naples, Italy; mariaida.maiorino@unicampania.it (M.I.M.); sergio.iorio@unicampania.it (S.I.); miriam.longo@unicampania.it (M.L.); giuseppe.bellastella@unicampania.it (G.B.); 2Division of Thyroid Surgery, University of Campania “L. Vanvitelli”, 80138 Naples, Italy; giovanni.docimo@unicampania.it; 3Department of Woman, Child, General and Specialized Surgery, University of Campania “L. Vanvitelli”, 80138 Naples, Italy; anna.grandone@unicampania.it (A.G.); caterina.luongo@unicampania.it (C.L.); emanuele.miragliadelgiudice@unicampania.it (E.M.D.G.); 4Pathology Unit, Department of Mental and Physical Health and Preventive Medicine, University of Campania “L. Vanvitelli”, 80138 Naples, Italy; immacolata.cozzolino@unicampania.it; 5Department of Nuclear Medicine, E.O. Ospedali Galliera, 16128 Genoa, Italy; arnoldo.piccardo@galliera.it; 6Clinic for Endocrinology and Diabetology, Regional Hospital of Lugano, Ente Ospedaliero Cantonale, 6900 Lugano, Switzerland; pierpaolo.trimboli@eoc.ch; 7Department of Medical and Advanced Surgical Sciences, University of Campania “L. Vanvitelli”, 80138 Naples, Italy; katherine.esposito@unicampania.it

**Keywords:** pediatric thyroid nodules, neck ultrasound

## Abstract

**Simple Summary:**

Although pediatric thyroid nodules are uncommon, they need high clinical expertise and alert since they carry a greater risk of malignancy compared with those presenting in adults. Since there are no specific ultrasound (US)-based risk stratification systems (RSSs) for pediatric thyroid nodules, the application of adult-based RSSs in the pediatric population could represent a step forward in the care of children and adolescents with thyroid nodules. We compared the diagnostic performance of the main US-based RSSs *i.e., the American College of Radiology (ACR), European (EU), Korean (K) Thyroid Imaging Reporting and Data Systems (TI-RADSs) and ATA US RSS criteria) for detecting malignant thyroid lesions in pediatric patients. For ACR TI-RADS and EU-TIRADS, we found a sensitivity of 41.7%, and, for K-TIRADS and ATA US RSS, we found a sensitivity of 50%. The four US-based RSSs (i.e., ACR-TIRADS, EU-TIRADS, K-TIRADS, and ATA US RSS) have suboptimal performance in managing pediatric patients with thyroid nodules, with one-half of cancers without indication for FNA according to their recommendations. All thyroidologists, as well as the panelists of next TIRADSs, should be aware of these findings.

**Abstract:**

Neck ultrasound (nUS) is the cornerstone of clinical management of thyroid nodules in pediatric patients, as well as adults. The current study was carried out to explore and compare the diagnostic performance of the main US-based risk stratification systems (RSSs) (i.e., the American College of Radiology (ACR), European (EU), Korean (K) TI-RADSs and ATA US RSS criteria) for detecting malignant thyroid lesions in pediatric patients. We conducted a retrospective analysis of consecutive children and adolescents who received a diagnosis of thyroid nodule. We included subjects with age <19 years having thyroid nodules with benign cytology/histology or final histological diagnosis. We excluded subjects with (a) a previous malignancy, (b) a history of radiation exposure, (c) cancer genetic susceptibility syndromes, (d) lymph nodes suspicious for metastases of thyroid cancer at nUS, (e) a family history of thyroid cancer, or (f) cytologically indeterminate nodules without histology and nodules with inadequate cytology. We included 41 nodules in 36 patients with median age 15 years (11–17 years). Of the 41 thyroid nodules, 29 (70.7%) were benign and 12 (29.3%) were malignant. For both ACR TI-RADS and EU-TIRADS, we found a sensitivity of 41.7%. Instead, for both K-TIRADS and ATA US RSS, we found a sensitivity of 50%. The missed malignancy rate for ACR-TIRADS and EU-TIRADS was 58.3%, while that for K-TIRADS and ATA US RSS was 50%. The unnecessary FNA prevalence for ACR TI-RADS and EU-TIRADS was 58.3%, while that for K-TIRADS and ATA US RSS was 76%. Our findings suggest that the four US-based RSSs (i.e., ACR-TIRADS, EU-TIRADS, K-TIRADS, and ATA US RSS) have suboptimal performance in managing pediatric patients with thyroid nodules, with one-half of cancers without indication for FNA according to their recommendations.

## 1. Introduction

Compared with those of adults, pediatric thyroid nodules have molecular and pathological peculiarities that promoted the development of unique pediatric guidelines [1,2,3]. The prevalence of ultrasound-detected thyroid nodules varies from 0.5% [4] to 1.6% [5] in the child population. Although pediatric thyroid nodules are uncommon, they need high clinical expertise and alert since they carry a greater risk of malignancy compared with those presenting in adults (22–26% versus 5–10%) [1,6,7]. Moreover, children with thyroid cancer are more likely than adults to have cervical lymph node metastases, extrathyroidal extension, and pulmonary metastases at the time of diagnosis, as well as persistence/recurrence of disease [1].

Neck ultrasound (nUS) is the cornerstone of the clinical management of thyroid nodule in pediatric patients, as well as adults [8,9,10,11,12]. According to the American Thyroid Association (ATA) guidelines [1], sonographic evaluation of thyroid nodules in children should be modeled on 2009 ATA guidelines for adults [13]. However, when exploring thyroid nodules at nUS in childhood, some peculiar aspects should be kept in mind: first, the fact that the size is a rather questionable parameter in children because thyroid volume changes with age; second, increased intranodular vascularity is apparently more common in malignant nodules; third, a diffusely infiltrative form of papillary thyroid cancer (PTC) is relatively frequent; fourth, the clinical context is of paramount importance when interpreting sonographic features [1,14,15]. 

US-based risk stratification systems (RSSs), often referred to as Thyroid Imaging Reporting and Data Systems (TIRADSs), have been developed to establish a standard lexicon to describe thyroid nodules, to associate nodules with a malignancy risk class, and to detect malignant nodules requiring fine-needle aspiration (FNA) [16]. RSSs mainly apply to PTC [17], since they have a lower performance in the detection of follicular thyroid carcinoma [18], medullary thyroid carcinoma [19,20], anaplastic thyroid carcinoma [21], and autonomously functioning thyroid nodules [22]. Moreover, RSSs have been extensively validated in the adult population [23], but not in children [24] and older adults [25]. Yet, the clinical context of patients is not considered in RSSs, and whether a patient’s age can modify their reliability is a matter of debate [25]. 

Since single thyroid US features are not highly accurate predictors of benign or malignant etiology of thyroid nodules in children, and specific RSSs for pediatric thyroid nodules are lacking, the application of adult-based RSSs in the pediatric population could represent a step forward in the care of children and adolescents with thyroid nodules [6,8]. Specifically, exploring the reliability of RSSs in the management of pediatric nodules could serve to create standardized diagnostic algorithms for childhood aimed at increasing our ability to detect thyroid cancer early.

To our knowledge, few studies [11,24] evaluated the diagnostic performance of RSSs in malignancy risk stratification of pediatric thyroid nodules with discordant results, and further studies on this topic are mandatory [24].

Therefore, the current study was carried out to explore and compare the diagnostic performance of the main RSSs (i.e., the American College of Radiology (ACR) [26], European (EU) [27], Korean (K) [28] TI-RADSs and ATA US RSS criteria [29]) for detecting malignant thyroid lesions in pediatric patients, in terms of risk stratification and reliability in the indication for FNA.

## 2. Materials and Methods

### 2.1. Study Design and Patients

In the current study, the Standards for Reporting Diagnostic Accuracy (STARD) statement was followed [30]. Specifically, we conducted a retrospective analysis of consecutive children and adolescents who received a diagnosis of thyroid nodule at a single referral center (i.e., Division of Endocrinology and Metabolic Diseases, University of Campania “L. Vanvitelli”—Naples, Italy) from January 2017 to March 2021. We gathered information (i.e., demographic, laboratory, imaging, and pathological details) from medical records included in the hospital database of pediatric patients referred to our multidisciplinary team since they developed clinical manifestations suspicious for hypothyroidism or thyrotoxicosis or were investigated for palpable thyroid nodules. 

Subjects fulfilling the following criteria were enrolled in the current study: (a) age <19 years; (b) thyroid nodule(s) with benign cytology/histology or final histological diagnosis (i.e., benignity or malignancy); (c) complete data (i.e., hormonal and antibodies profile including serum calcitonin; at least two clear B-Mode US images for each nodule). 

Patients were excluded if they had (a) a previous malignancy, (b) a history of radiation exposure, (c) cancer genetic susceptibility syndromes, (d) lymph nodes suspicious for metastases of thyroid cancer at nUS, (e) a family history of thyroid cancer (i.e., at least one relative), or (f) cytologically indeterminate nodules without histology and nodules showing inadequate cytology. 

The Ethics Committee of the University Hospital “L. Vanvitelli” (Naples, Italy) approved the study, and written consent was obtained from all the participants.

### 2.2. Thyroid Ultrasonography

Thyroid ultrasonography was performed by the same experienced operator (S.I.) using an ultrasound device (MyLab^TM^Six, Esaote) with a 7–14 MHz wide-band linear transducer. The color gain was adjusted so that artefacts were prevented. The examination of ultrasonographic features of thyroid nodules, along with thyroid vascularity and volume, were systematically conducted for patients presenting for thyroid assessment to our division.

In the current study, US images were reviewed, and ACR-TIRADS, EU-TIRADS, K-TIRADS, and ATA US RSS criteria were applied to each nodule for categorization separately by two experienced thyroidologists (L.S., G.B.) who were unaware of the nodule’s cytopathology and histopathology, as well as of laboratory and imaging results. In the case of discordant US categorization, a consensus with the help of a third reviewer (P.T.) (also unaware of pathology or any other patient data) was reached.

### 2.3. Thyroid Nodule Pathology

In the Division of Anatomic Pathology of our institution, all FNAs were reported according to the revised Italian Consensus for the Classification and Reporting of Thyroid Cytology [31] and the final pathology (i.e., histology of the thyroid nodule after surgery) according to the World Health Organization (WHO) book on endocrine tumor classification [32]. For our pediatric thyroid nodules, benignity at cytological or histological exam and malignancy at histopathology were reference standard for the calculation of the diagnostic performance of US RSSs. Indication to perform FNA of thyroid nodules was made by the clinician (i.e., endocrinologist or pediatrician) according to US features, laboratory, other imaging (i.e., scintigraphy, if necessary), individual risk of malignancy, and patient/family preference. Indeterminate nodules at cytology often underwent surgery, or they were followed up on the basis of in-house molecular testing results, US features, and patient/family preference.

### 2.4. Statistical Analysis

Continuous variables were described as median and interquartile range (IQR). Categorical variables were presented as number (percentage). The diagnostic performance of the main RSSs was expressed through predictivity tests (i.e., sensitivity, specificity, positive (PPV) and negative (NPV) predictive value, and accuracy, with specific 95% confidence intervals), which were calculated according to Galen and Gambino [33], and the unnecessary FNA prevalence, defined as the number of benign nodules among the FNA-required nodules. Specifically, we employed assessments of malignant versus benign nodules in order to be able to report the estimates of accuracy on a lesion basis.

The interobserver agreement was evaluated by Cohen’s kappa statistic, where the kappa value (*k*) denotes the strength of agreement and is interpreted as follows: 0–0.2, poor; 0.2–0.4, fair; 0.4–0.6, moderate; 0.6–0.8, good; 0.8–1.0 very good. Statistical significance was defined as a *p*-value < 0.05. Statistical analysis was performed by MedCalc software version 9 (Mariakerke, Belgium).

## 3. Results

There were 81 thyroid nodules in 71 patients undergoing both nUS and FNA in the initial database. After applying our exclusion criteria, in our study, we finally included 41 nodules in 36 patients (Figure 1). Twenty-six patients were female (72.2%), and ten patients were male (27.8%). Median age was 15 years (11–17 years), with the final cohort including 12 prepubertal and 24 postpubertal patients. The nUS indication was the following: autoimmune chronic thyroiditis (±hypothyroidism) in 17 patients (47.2%); excluding thyroid disease in nine patients (25.0%); palpable thyroid nodules (±goiter) in six patients (16.7%); Graves’ hyperthyroidism in four patients (11.1%). Among the 36 patients, 28 (77.8%) had a solitary thyroid nodule, and eight patients (22.2%) had multiple thyroid nodules. The median nodule’ s maximal dimension was 13 mm (10–16 mm). Of the 41 thyroid nodules, 29 (70.7%) were benign (of which six (20.7%) underwent surgery) and 12 (29.3%) were malignant. Serum calcitonin was negative in all cases.

Most cancers were papillary carcinoma (10 (83.3%) of 12, nine conventional variants, including one multifocal and one follicular variant), followed by follicular carcinoma (two (16.7%) of 12). Median maximal dimension of malignant thyroid nodules was 10 mm (7–13). The characteristics of our patients are shown in Table 1.

The distribution of thyroid nodules according to the ACR-TIRADS, EU-TIRADS, K-TIRADS, and ATA US RSS risk levels is summarized in Table 2. The highest number of nodules (16 of 41 nodules) fell into the intermediate-risk category (i.e., TR4, EU-TIRADS 4, K-TIRADS 4, intermediate suspicion). A 100% cancer prevalence was observed in the high-risk class (i.e., TR5, EU-TIRADS 5, K-TIRADS 5, high suspicion). While 6/12 (50%) of cancers were assessed by the highest-risk category (i.e., TR5, EU-TIRADS 5, K-TIRADS 5, high suspicion), the remaining half were classified as at low or intermediate risk in all US RSSs.

Table 3 shows the recommended management of nodules in this cohort based on the ACR-TIRADS, EU-TIRADS, K-TIRADS, and ATA US RSS criteria. Among 29 benign nodules, according to ACR TI-RADS and EU-TIRADS criteria, seven (24.1%) would have undergone FNA, while 22 (75.9%) would have been followed up without FNA. Instead, among 29 benign nodules, according to K-TIRADS and ATA US RSS criteria, 19 (65.5%) would have undergone FNA, while 10 (34.5%) would have been followed up without FNA. According to the ACR TI-RADS and EU-TIRADS criteria, five (41.7%) of the 12 malignant nodules would have undergone FNA, and seven (58.3%) would have been assigned follow-up without FNA. According to the K-TIRADS and ATA US RSS criteria, six (50%) of the 12 malignant nodules would have undergone FNA, and six (50%) would have been assigned follow-up without FNA. The unnecessary FNA prevalence for ACR TI-RADS and EU-TIRADS was 58.3%, while that for K-TIRADS and ATA US RSS was 76%. 

Table 4 resumes the reliability of the four RRSs in correctly indicating FNA. Specifically, for ACR TI-RADS and EU-TIRADS, we found the following results: sensitivity 41.7%, specificity 75.9%, PPV 41.7%, NPV 75.9%, and accuracy 65.9%. Instead, for K-TIRADS and ATA US RSS, we found the following results: sensitivity 50%, specificity 34.5%, PPV 24%, NPV 62.5%, and accuracy 39%. The interobserver agreement in classifying nodules according to ACR-TIRADS, EU-TIRADS, K-TIRADS, and ATA US RSS was good with *k*-values of 0.7, 0.61, 0.66, and 0.62, respectively (*p* ≤ 0.002 in all cases).

The clinical and US characteristics of malignant nodules that would not have undergone FNA according to the RRSs are presented in Table 5. Six nodules (ID 1, 2, 4, 5, 6, and 7) would not have been identified using the four RSSs at initial visit, while one nodule (ID 3) would have undergone FNA according to K-TIRADS and ATA US RSS but not according to ACR-TIRADS and EU-TIRADS criteria. The median maximal dimension of these seven malignant nodules was 10 mm (7–12 mm). Five of the seven nodules were solitary. These were papillary carcinoma in five cases (four with conventional variant, including one multifocal and one follicular variant) and follicular carcinoma in two cases.

## 4. Discussion

### 4.1. Principal Findings

In our final pediatric cohort, we found a malignancy rate (nearly 30%) similar to that reported in previous studies of children and higher than that associated with thyroid nodules in adults [1,6,7]. The risk of malignancy was highest for the high-risk levels of all four RSSs (i.e., TR5, EU-TIRADS 5, K-TIRADS 5, high suspiscion) with 100% concordance between the US-based high-risk level and cancer. This finding was roughly in line with the adult-based estimated risk of malignancy reported in the four RSSs (i.e., >20% ACR-TIRADS, 26–87% EU-TIRADS, >60% K-TIRADS, and >20% ATA US RSS) [26,27,28,29]. Likewise, regarding the intermediate-risk level (i.e., TR4, EU-TIRADS 4, K-TIRADS 4, intermediate suspiscion), we found that the malignancy rate of 12.5% for all four RSSs was comparable to that reported in adults (i.e., 5–20% ACR-TIRADS, 6–17% EU-TIRADS, 15–50% K-TIRADS, and 10–20% ATA US RSS) [26,27,28,29]. Conversely, we found a risk of malignancy of about 30% associated with low-risk levels of all four RSSs (i.e., TR3, EU-TIRADS 3, K-TIRADS 3, low suspicion), which was relevant to and higher than that reported in adults (i.e., 5% ACR-TIRADS, 2–4% EU-TIRADS, 3–15% K-TIRADS, and 5–10% ATA US RSS) [26,27,28,29]. Moreover, a non-negligible risk of malignancy of 12.5–20% was associated with not suspicious/benign risk levels for ACR-TIRADS, EU-TIRADS, and K-TIRADS (i.e., TR2, EU-TIRADS 2, K-TIRADS 2), which was higher than the rates of adults (i.e., <2% ACR-TIRADS, 0% EU-TIRADS, and 1–3% K-TIRADS) [26,27,28,29]. All this means that, compared to thyroid nodules in adults, the probability of finding cancer in high- and intermediate-risk levels of the four RSSs (i.e., ACR-TIRADS, EU-TIRADS, K-TIRADS, and ATA US RSS) remains high and is not negligible for not suspicious/benign risk levels per ACR-TIRADS, EU-TIRADS, and K-TIRADS. These results are in line with what emerged in large studies by Richman et al. [34], Lee et al. [35], and Martinez-Rios et al. [36], where a significant number of malignant nodules fell in low-risk RSS categories.

While the majority of cancers (8/12, 66.7%) in our study fell within high- and intermediate-risk categories per all the four RSSs, as resumed in Table 5, six of the 12 cancers (50%) would not have undergone FNA at the initial visit according to all the four RSSs. One more cancer (ID 3) with a maximum dimension of 13 mm and intermediate-risk category would not have undergone FNA according to ACR-TIRADS and EU-TIRADS criteria. Two PTCs scored as high-risk lesions per all four RSSs would not have undergone FNA since they were 7 mm of maximum dimension. One PTC (ID 2) and the two FTCs (ID 5, ID 6) of the present cohort were scored as low-risk lesions, and, because of their size (<15 mm), FNA would have not been indicated per all four RSSs. The remaining PTC (ID 4) with maximum dimension <20 mm (i.e., 7 mm) fell within the not suspicious/benign risk categories per ACR-TIRADS, EU-TIRADS, and K-TIRADS (i.e., TR2, EU-TIRADS 2, K-TIRADS 2) and low-risk category per ATA US RSS; thus, it would not have undergone FNA. 

Therefore, a high missed malignancy rate (~50%) was found in our study when using ACR-TIRADS, EU-TIRADS, K-TIRADS, and ATA US RSS. This result is conceptually comparable to what was reported by the largest study by Richman et al. [34], who found a 22.1% of missed malignancy rate applying ACR-TIRADS, and by Lee et al. [35], who found a 19.2% of missed malignancy rate applying K-TIRADS in the group without risk factors. This issue likely implies that the current RSSs (i.e., ACR-TIRADS, EU-TIRADS, K-TIRADS, and ATA US RSS) are likely inadequate for guiding FNA of thyroid nodules in patients younger than 19 years old. In this regard, as already shown for thyroid nodules in adults [19,20], we can hypothesize that the presence of two FTC cases in our pediatric cohort would also have increased the missed malignancy rate and, thus, decreased the overall ability of the four RSSs in detecting malignant nodules. Although a direct comparison with the adult population is somewhat difficult because the missed malignancy rate is largely influenced by the proportion of malignant nodules, we found the missed malignancy rate for the four RSSs to be significantly higher than that reported in the literature relative to adult patients (i.e., 2.2–9.5%) [24,37]. While this evidence may be acceptable for adult with thyroid nodules where US-based risk stratification systems are now mainly applied to detect clinically important cancers and to avoid waste of resources (conservative approach), this may not be applied in children and adolescents where the first aim should consist of early detection of malignant nodules.

One other parameter underlying the diagnostic performance of RSSs is represented by the unnecessary FNA rate. For management of thyroid nodules in children and adolescents, this parameter could be less important to improve, as the primary objective is detecting malignancy. However, we found higher unnecessary FNA rates (i.e., almost 60% for ACR-TIRADS and EU-TIRADS, and almost 80% for K-TIRADS and ATA US RSS) than recently reported by Kim et al. [38] for adults (pooled unnecessary FNA rates of ACR-TIRADS, EU-TIRADS, K-TIRADS, and ATA were 25%, 38%, 55%, and 51%, respectively). The higher unnecessary FNA rates of K-TIRADS/ATA US RSS than ACR-TIRADS/EU-TIRADS could also be due to the lower cutoffs for FNA associated with intermediate- and low-risk categories (i.e., 10 mm and 15 mm, respectively, compared to 15 and 20–25 mm of ACR-TIRADS and EU-TIRADS) [26,27,28,29].

In our cohort, the overall accuracy of the four RSSs in correctly indicating FNA was quite poor (i.e., ~66% for ACR-TIRADS and EU-TIRADS, ~40% for K-TIRADS and ATA US RSS). In particular, sensitivity values (i.e., 40–50%), although slightly higher for K-TIRADS and ATA US RSS, were inadequate to properly detect malignancy in this context, and they were significantly lower than that reported in adults (74% ACR-TIRADS, 54% EU-TIRADS, 86% K-TIRADS, and 87% ATA US RSS) [23]. 

All this suggests that, on the one hand, the four RSSs had an excellent yield in high-risk US nodules but, on the other hand, they should be appropriately modified to detect the best number of malignancies in children.

### 4.2. Strengths and Weaknesses

The strengths of our study are the following: (1) to our knowledge, this is the first comparative study regarding diagnostic performance of the four most used RSSs for detecting malignant thyroid lesions in pediatric patients; (2) in comparison with the largest study to date by Richman et al. [31], this study mainly provides data for the management of small thyroid nodules and cancers (the median nodule’s maximal dimension was 13 mm, and the median maximal dimension of malignant thyroid nodules was 10 mm). The limitations of our study should also be discussed. This was a small and monocentric cohort. However, we strictly selected the cohort by excluding patients with apparent risk factors of malignancy, so that our results could be mainly applied to the majority of children and adolescents with sporadic thyroid cancer. Although we included patients with preexisting autoimmune thyroid disease, the putative role of the autoimmune background in the development of thyroid cancer in childhood is inconclusive to date [39,40]. This is a retrospective review of static US images which could result in inherent selection bias by the reviewers. However, interobserver agreement in scoring nodules according to all four RSSs was good. Patients with benign cytology could undergo surgical resection in the future, altering the current results of the current study. However, one-fifth of our benign cases received surgery and had histological confirmation. Since we did not have complete data on nUS relative to the vascularity of thyroid nodules, we could not assess this feature and score our nodules according to AACE/ACE/AME US RSS [41]. Our results mainly refer to PTC without apparent risk factors.

## 5. Conclusions

Our findings suggest that the four US-based RSSs (i.e., ACR-TIRADS, EU-TIRADS, K-TIRADS, and ATA US RSS) have suboptimal performance in managing pediatric patients with thyroid nodules, with one-half of cancers being without indication for FNA according to their recommendations. All thyroidologists, endocrinologists, and radiologists, as well as panelists of later TIRADSs, should be aware of these findings [42].

## Figures and Tables

**Figure 1 cancers-13-05304-f001:**
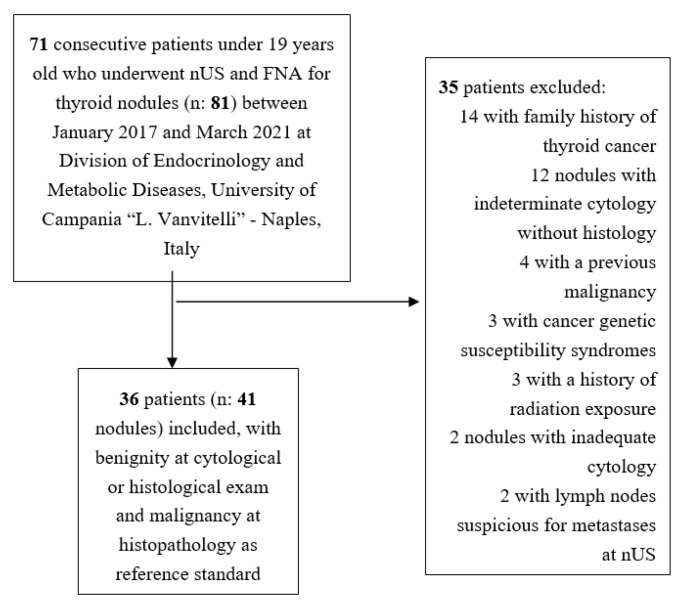
Flowchart of patient selection. nUS, neck ultrasound; FNA, fine-needle aspiration.

**Table 1 cancers-13-05304-t001:** Main characteristics of our patients (*n* = 36).

Characteristics	
Age at diagnosis, years (IQR)	15 (11–17)
Females/males (*n*)	26/10
Reasons to perform nUS	
autoimmune chronic thyroiditis, *n* (%)	17 (47.2)
no thyroid disease, *n* (%)	9 (25)
palpable thyroid nodules, *n* (%)	6 (16.7)
Graves’ hyperthyroidism, *n* (%)	4 (11.1)
Nodules	
maximal dimension, mm (IQR)	13 (10–16)
solitary, *n* (%)	28 (77.8)
multiple, *n* (%)	8 (22.2)
Benign nodules/malignant nodules, *n* (%)	29/12 (70.7/29.3)
benign with surgery Malignant nodules,	6/29 (20.7)
maximal dimension, mm (IQR)	10 (7–13)
PTC, *n* (%)	10 (83.3)
FTC, *n* (%)	2 (16.7)

IQR, interquartile range; nUS, neck ultrasound; mm, millimeter; PTC, papillary thyroid cancer; FTC, follicular thyroid cancer.

**Table 2 cancers-13-05304-t002:** Distribution of 41 thyroid nodules according to the ACR-TIRADS, EU-TIRADS, K-TIRADS, and ATA US RSS risk levels in 36 patients in our cohort.

ACR-TIRADS	Benign Nodules (*n*)	Malignant Nodules (*n*)	Total Nodules (*n*)	Cancer Prevalence (%)
TR1	4	0	4	0
TR2	4	1	5	20
TR3	7	3	10	30
TR4	14	2	16	12.5
TR5	0	6	6	100
**EU-TIRADS**				
2	7	1	8	12.5
3	8	3	11	27.3
4	14	2	16	12.5
5	0	6	6	100
**K-TIRADS**				
2	7	1	8	12.5
3	8	3	11	27.3
4	14	2	16	12.5
5	0	6	6	100
**ATA US RSS**				
benign	0	0	0	0
very low suspicion	5	0	5	0
low suspicion	10	4	14	28.6
intermediate suspicion	14	2	16	12.5
high suspicion	0	6	6	100

TIRADS, Thyroid Imaging Reporting and Data System; US RSS, ultrasound-based risk stratification system; ACR, American College of Radiology; EU, European; K, Korean; ATA, American Thyroid Association.

**Table 3 cancers-13-05304-t003:** Management of 41 thyroid nodules according to the the ACR-TIRADS, EU-TIRADS, K-TIRADS, and ATA US RSS criteria in 36 patients in our cohort.

Management Per ACR TIRADS Criteria	Benign Nodules (*n*)	Malignant Nodules (*n*)	Total Nodules (*n*)	Cancer Prevalence (%)	Unnecessary FNA Prevalence (%)
FNA	7	5	12	41.7	58.3
Follow-up/no FNA	22	7	29	24.1	
**Management per** **EU-TIRADS criteria**					
FNA	7	5	12	41.7	58.3
Follow-up/no FNA	22	7	29	24.1	
**Management per** **K-TIRADS criteria**					
FNA	19	6	25	24	76
Follow-up/no FNA	10	6	16	37.5	
**Management per** **ATA US RSS criteria**					
FNA	19	6	25	24	76
Follow-up/no FNA	10	6	16	37.5	

TIRADS, Thyroid Imaging Reporting and Data System; US RSS, ultrasound-based risk stratification system; ACR, American College of Radiology; EU, European; K, Korean; ATA, American Thyroid Association; FNA, fine-needle aspiration. The unnecessary FNA prevalence for the diagnosis of thyroid cancer was defined as the number of benign nodules among the FNA-required nodules.

**Table 4 cancers-13-05304-t004:** Reliability of the ACR-TIRADS, EU-TIRADS, K-TIRADS, and ATA US RSS in correctly indicating FNA in 41 nodules of 36 patients in our cohort.

	Sensitivity (%) (CI)	Specificity (%) (CI)	PPV (%) (CI)	NPV (%) (CI)	Accuracy (%)
**ACR TIRADS**	41.7 (27–58)	75.9 (60–87)	41.7 (27–58)	75.9 (60–87)	65.9
**EU-TIRADS**	41.7 (27–58)	75.9 (60–87)	41.7 (60–87)	75.9 (60–87)	65.9
**K-TIRADS**	50.0 (32–68)	34.5 (0.3–14)	24.0 (11–43)	62.5 (0.3–14)	39.0
**ATA US RSS**	50.0 (32–68)	34.5 (0.3–14)	24.0 (11–43)	62.5 (0.3–14)	39.0

TIRADS, Thyroid Imaging Reporting and Data System; US RSS, ultrasound-based risk stratification system; ACR, American College of Radiology; EU, European; K, Korean; ATA, American Thyroid Association; FNA, fine-needle aspiration; PPV, positive predictive value; NPV, negative predictive value; CI, 95% confidence interval.

**Table 5 cancers-13-05304-t005:** Clinical and US characteristics of proven malignancies not identified with the RSSs (ACR-TIRADS, EU-TIRADS, K-TIRADS, and ATA US RSS).

ID	Age (Years)	Gender	Number	Location	Composition	Echogenicity	Taller Than Wide	Margin	Echogenic Foci	Maximum Dimension (mm)	TI-RADS Risk Level	Cytology	Histology	Preexisting Thyroid Disease
**1**	17	F	single	lower left pole	solid	hypoechoic	no	ill-defined	punctate	7	TR5, EU5, K5, High	TIR4	mCPTC	no
**2**	15	M	multiple	mid right lobe	solid	isoechoic	no	smooth	no	10	TR3, EU3, K3, Low	TIR5	CPTC	no
**3**	18	F	single	isthmus	solid	hypoechoic	no	smooth	no	13	TR4, EU4	TIR3A	FV-PTC	ACT
**4**	17	F	single	upper right lobe	mixed cystic and solid	isoechoic	no	smooth	no	7	TR2, EU2, K2, Low	TIR5	CPTC	GD
**5**	7	M	multiple	mid right lobe	solid	isoechoic	no	smooth	no	10	TR3, EU3, K3, Low	TIR3B	FTC	ACT
**6**	12	M	single	upper left lobe	solid	isoechoic	no	smooth	no	12	TR3, EU3, K3, Low	TIR3B	FTC	ACT
**7**	13	F	single	upper right lobe	solid	hypoechoic	no	ill-defined	punctate	7	TR5, EU5, K5, High	TIR5	CPTC	ACT

TIRADS, Thyroid Imaging Reporting and Data System; US RSS, ultrasound-based risk stratification system; ACR, American College of Radiology; EU, European; K, Korean; ATA, American Thyroid Association; FNA, fine-needle aspiration; mCPTC, multifocal conventional papillary thyroid cancer; CPTC, conventional papillary thyroid cancer; ACT, autoimmune chronic thyroiditis; follicular variant of papillary thyroid cancer; GD, Graves’ disease; FTC, follicular thyroid cancer.

## Data Availability

The data presented in this study are available on request from the corresponding author.

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
