# Peer review of "Exploring the Performance of Ultrasound Risk Stratification Systems in Thyroid Nodules of Pediatric Patients"

_cancers, 2021, doi:10.3390/cancers13215304_

Round 1

Reviewer 1 Report

The authors present a review of pediatric thyroid nodules as evaluated using established risk stratification scales for the adult population

What features did the authors find to be the most predictive in the pediatric population?

Author Response

Authors: we greatly thank the reviewer for appreciating our work. As for adults, the most predictive features were nodular microcalcifications and irregular or lobulated margins. Indeed, as shown in Table 2, according to the four RSSs the six high risk nodules were malignant at the final histology.

Reviewer 2 Report

This is an interesting study on the use of RSS for diagnosis and prognostication of thyroid nodules in the pediatric population. It is a well written study. I do have a few minor comments:

  • the inclusion criterion b in line 115 is not clear. Please clarify. Were only the nodules that were benign on FNA included? I suppose not. I think malignancy in final pathology is missing
  • Please correct Figure 1. The text has been left off the boxes, and it does not make sense.
  • Why did the 6 benign nodules undergo surgery? Were they benign on FNA as well? If so, how big were they and why surgery?
  • I would like to see the final pathology for the cancers, what aggressive features they had. This is especially true for the nodules that were not supposed to be FNA'ed based on RSS. This is relevant as it would be crucial to the reader to see if the cancers we are missing can be very aggressive, or they can also be taken out at a later time.
  • Was there any association with the misclassification of nodules per the RSS and Graves' disease and/or Hashimoto's? 
  • In the discussion section, it seems a bit contradictory that there were a lot of missed malignancies, but also a lot of unnecessary FNAs. I think the authors should propose a solution. Should the RSS never be used in pediatric population? It seems to me that all RSS had an excellent yield in high US risk nodules. Also, the ATA seemed very good in low and very low risk nodules. The authors should discuss that and offer a potential solution to the readers.

Author Response

Comments to the Authors:

  1. “This is an interesting study on the use of RSS for diagnosis and prognostication of thyroid nodules in the pediatric population. It is a well written study. I do have a few minor comments:

the inclusion criterion b in line 115 is not clear. Please clarify. Were only the nodules that were benign on FNA included? I suppose not. I think malignancy in final pathology is missing”

Authors: we thank the reviewer to critically revise our study. We thought it was included in “final histological diagnosis”. However, accordingly we added “malignancy in final pathology” in lines 115-116.

  1. “Please correct Figure 1. The text has been left off the boxes, and it does not make sense.”

Authors: we greatly thank the reviewer for this remark. We do not know how it happened, since figure 1 was correctly uploaded. Maybe the automatic conversion by the journal from Arial to TimesNewRoman made these errors. Now, we directly corrected the figure 1 in the main text sent us by Cancers.

  1. “Why did the 6 benign nodules undergo surgery? Were they benign on FNA as well? If so, how big were they and why surgery?.”

Authors: we thank the reviewer for this relevant remark. The six histologically benign nodules were 1 TIR2 (benign), 3 TIR3A (low-risk indeterminate), 2 TIR3B (high-risk indeterminate). The patient with TIR2 was operated because she had goiter associated to inestetism and dysphagia. The remaining 5 indeterminate cases followed surgery because of RAS positivity (1TIR3B h-RAS + and 1 TIR3A n-RAS +), goiter (1TIR3A), not negligible risk of malignancy and preferred by the patient (1TIR3A wild), high-risk of malignancy (1 TIR3B).

  1. “I would like to see the final pathology for the cancers, what aggressive features they had. This is especially true for the nodules that were not supposed to be FNA'ed based on RSS. This is relevant as it would be crucial to the reader to see if the cancers we are missing can be very aggressive, or they can also be taken out at a later time.

Authors: we thank the reviewer for this relevant suggestion. Table 5 give the details of the seven malignant nodules that would not have undergone FNA according to the RRSs. They were 4 conventional PTC, 1 follicular PTC, 2 FTC. Thus, in this series biological aggressiveness is theoretically heterogeneous.

  1. “Was there any association with the misclassification of nodules per the RSS and Graves' disease and/or Hashimoto's?.”

Authors: we thank the reviewer for this important remark. Statistical association between the misclassification and autoimmune thyroid diseases is not applicable due to the small cohort of our patients and of cancers. However, in at least 3 malignant cases (see Table 5, ID 4, 5, 6) that would have not undergone FNA the US appearance of background parenchima could have altered the features suggestive of malignancy and/or their interpretation at US imaging.

  1. “In the discussion section, it seems a bit contradictory that there were a lot of missed malignancies, but also a lot of unnecessary FNAs. I think the authors should propose a solution. Should the RSS never be used in pediatric population? It seems to me that all RSS had an excellent yield in high US risk nodules. Also, the ATA seemed very good in low and very low risk nodules. The authors should discuss that and offer a potential solution to the readers.”

Authors: we thank the reviewer for this important suggestion. Accordingly, in the discussion we added the following sentence: “All this suggests that on the one hand the four RSSs had an excellent yield in high US risk nodules but, on the other hand, they should be appropriately modified to detect the best number of malignancies in children”.

However, ATA had the same characteristics of accuracy as K-TIRADS. Moreover while both ATA and K-TIRADS missed 6 cancers out of 12 (one cancer less thanks to the higher propensity to indicate FNA), EU- and ACR-TIRADS missed 7 cancers out of 12.

Reviewer 3 Report

The manuscript entitled “Exploring the performance of ultrasound risk stratification systems in thyroid nodules of pediatric patients" reports a very interesting and important clinical problem.

However this research has serious flaws.

  1. The number of cases was extremely small for this type of research (36 patients/41 nodules diagnosed/cured between 01.2017-03.2021), and further studies with larger sample sizes are necessary.
  2. This was a single-center, retrospective study, which may have reduced the statistical significance.
  3. Median patients’ age was 15 years (11-17 years) – it is more of young adults’ group than children.
  4. The conclusion is not very convincing, because of main limitation of this research.

Author Response

he manuscript entitled “Exploring the performance of ultrasound risk stratification systems in thyroid nodules of pediatric patients" reports a very interesting and important clinical problem.

However this research has serious flaws.

  1. The number of cases was extremely small for this type of research (36 patients/41 nodules diagnosed/cured between 01.2017-03.2021), and further studies with larger sample sizes are necessary.

Authors.: we thank the reviewer for this remark. We agree that larger studies are necessary to derive solid conclusions. However, our small cohort included higly selected patients,

  1. This was a single-center, retrospective study, which may have reduced the statistical significance.

Authors: we agree with the reviewer. Indeed this issue was considered as a limitation

  1. Median patients’ age was 15 years (11-17 years) – it is more of young adults’ group than children.

Authors: we thank the reviewer for this remark. Unfortunately under 15 and under 11 were less represented in our cohort.

  1. The conclusion is not very convincing, because of main limitation of this research.

Authors: although based on a small cohort, conclusions derived from results.

Reviewer 4 Report

The single-center, comparative, retrospective study (comprising data from January 2017 to March 2021) ”Exploring the performance of ultrasound risk stratification systems in thyroid nodules of pediatric patients” by Scappaticcio and colleagues set out to investigate the potential of adult-based risk stratification systems (RSSs) in pediatric patients as there currently are not specific ultrasound-RSSs for pediatric thyroid nodules. The study was based on 36 patients contributing with 41 nodules. The data support the finding that the four ultrasound-based RSSs from the USA, Europe, and Korea have suboptimal performance in managing pediatric patients with thyroid nodules; however, the reporting needs further improvement (see below). Apart from that, the paper is clearly written, easy to follow, and transparent in what has been done. Applicable reporting standards (STARD) have been applied.

I wonder whether the exclusion criteria were too strict. Why have patients with lymph nodes suspicious for metastases of thyroid cancer at neck ultrasound (exclusion criterion d) been excluded? Why was family history of thyroid cancer (i.e., at least one relative) a knock-out criterion (e)?

How did cytologically indeterminate nodules without histology and nodules showing inadequate cytology turn out for the patients. Were these nonmalignant after follow-up?

For estimates of sensitivity, specificity, positive and negative predictive values, and accuracy, respective 95% confidence intervals MUST be given (Table 4); without 95% CIs, the uncertainty with which the accuracy parameters are estimated is not quantified.

In section 2.4 (Statistical analysis), there is just a reference to the textbook by Galen & Gambino (1975). Instead of that reference, the authors should specify briefly how true positive, false negative, true negative, and false negative assessments were arrived at (what were the criteria). Especially, the authors need to underline that the analyses were lesion-based (in opposition to patient-based), and I guess the authors employed assessments of malignant versus benign nodules in order to be able to report specificity and negative predictive value on a lesion-basis.

The interobserver reliability was actually ‘good’ (line 237, also line 346) according to the criteria laid out in the Methods sections (line 159) as kappa exceeded 0.6.

The discussion would benefit from structuring. Docherty and Smith made such a proposal in an earlier BMJ editorial (DOI: 10.1136/bmj.318.7193.1224); the headlines can, of course, be adjusted, especially in view of this study most likely being the first of its kind comparing the four RSSs.

Line 160: What does ‘(Mariakerke)’ mean?

Author Response

Comments to the Authors:

  1. “The single-center, comparative, retrospective study (comprising data from January 2017 to March 2021) ”Exploring the performance of ultrasound risk stratification systems in thyroid nodules of pediatric patients” by Scappaticcio and colleagues set out to investigate the potential of adult-based risk stratification systems (RSSs) in pediatric patients as there currently are not specific ultrasound-RSSs for pediatric thyroid nodules. The study was based on 36 patients contributing with 41 nodules. The data support the finding that the four ultrasound-based RSSs from the USA, Europe, and Korea have suboptimal performance in managing pediatric patients with thyroid nodules; however, the reporting needs further improvement (see below). Apart from that, the paper is clearly written, easy to follow, and transparent in what has been done. Applicable reporting standards (STARD) have been applied.

I wonder whether the exclusion criteria were too strict. Why have patients with lymph nodes suspicious for metastases of thyroid cancer at neck ultrasound (exclusion criterion d) been excluded? Why was family history of thyroid cancer (i.e., at least one relative) a knock-out criterion (e)?

Author: we greatly thank the reviewer for appreciating our work.

We decided to make the exclusion criteria more strict (excluding 2 patients with metastatic lymph nodes and 14 patients with family history of thyroid cancer) to derive conclusions to be strictly applied to nodules and their features (and not to factors routinely to be considered to perform FNA such as family history and lymph nodes suspicious at nUS).

  1. “How did cytologically indeterminate nodules without histology and nodules showing inadequate cytology turn out for the patients. Were these nonmalignant after follow-up?”

Authors: We thank the reviewer for this remark. Unfortunately, we do not have final data regarding the nature (i.e., malignancy or begignitiy) of the nodules both for the 12 nodules with indeterminate cytology without histology and the  2 nodules with inadequate cytology .

  1. For estimates of sensitivity, specificity, positive and negative predictive values, and accuracy, respective 95% confidence intervals MUST be given (Table 4); without 95% CIs, the uncertainty with which the accuracy parameters are estimated is not quantified.

Authors: We thank the reviewer for this remark. In Table 4 the 95% confidence interval was added for each estimate of diagnostic accuracy. Moreover, in statical analysis was declared this change.

  1. In section 2.4 (Statistical analysis), there is just a reference to the textbook by Galen & Gambino (1975). Instead of that reference, the authors should specify briefly how true positive, false negative, true negative, and false negative assessments were arrived at (what were the criteria). Especially, the authors need to underline that the analyses were lesion-based (in opposition to patient-based), and I guess the authors employed assessments of malignant versus benign nodules in order to be able to report specificity and negative predictive value on a lesion-basis.

Authors: We thank the reviewer for this important remark. In statistical analysis we specified the above suggestions.

  1. The interobserver reliability was actually ‘good’ (line 237, also line 346) according to the criteria laid out in the Methods sections (line 159) as kappa exceeded 0.6.

Authors: We thank the reviewer for this remark. Ye, we agree it was good and we modified the text (results and discussion.

  1. The discussion would benefit from structuring. Docherty and Smith made such a proposal in an earlier BMJ editorial (DOI: 10.1136/bmj.318.7193.1224); the headlines can, of course, be adjusted, especially in view of this study most likely being the first of its kind comparing the four RSSs.

Authors: We thank the reviewer for this suggestion. We structured the discussion and the conclusions as suggested by DOI: 10.1136/bmj.318.7193.1224, also by creating some new paragraphs.

  1. Line 160: What does ‘(Mariakerke)’ mean?

Authors: We thank the reviewer for this remark. Mariakerke is in Belgium where the software was created. We added “Belgium” close to Mariakerke.

Reviewer 5 Report

Dear authors,

With great interest I read the manuscript “Exploring the performance of ultrasound risk stratification systems in thyroid nodules of pediatric patients” submitted to Cancers. The study retrospectively evaluates the diagnostic performance of the main risk stratification systems for detecting malignant thyroid nodules in a study cohort of 36 under age patients with 41 thyroid nodules.

The submitted manuscript is well designed and its methodology and statistical analysis are appropriate for its stated aim. The intention for the study is presented adequately and the results are clear-cut. It is well known, that pediatric thyroid nodules need a high clinical expertise for their higher malignancy risk compared to adults. Unfortunately, there are only few studies in current international literature evaluating diagnostic performance of the main risk stratification systems for detecting malignant thyroid nodules in children. The results of the submitted manuscript are interesting especially for endocrinologists, surgeons, radiologists and nuclear medicine physicians and therefore certainly worth publication. The results may also be a valuable help for the development of upcoming guidelines.

With the presented study, the authors did high quality research. However, I have minor revisions concerning the manuscript.

Introduction

The introduction is clearly to the point and the problem is described adequately.

Materials & Methods

This section is well structured and covers all essential information.

Results

I don’t know if I got the following point right: You included 36 patients with 41 thyroid nodules (what is consistent in the manuscript). In the results section you wrote that “Among the 36 patients, 28 had a solitary thyroid nodule and eight had multiple thyroid nodules”. So if “multiple nodules” means more than 1 nodule there should be at least 16 nodules in these 8 patients. Added to the 28 solitary nodules this would be a total of 44 nodules? Please clarify this point.

In figure 1 the text is not complete (at least in my pdf version). The upper box ends with “University of …??”, the lower box ends with “…at histopathology as ..:??” and the right box listing the excluded patients ends with “..nodules with inadequate…??”.

Moreover, there are 35 patients excluded in figure 1: 14 with family history of thyroid cancer, 12 with indeterminate cytology without histology, 4 with previous malignancy, 3 with cancer genetic susceptibility syndroms, 3 with a history of radiation exposure and 2 nodules with inadequate??. 14+12+4+3+3+2 is 38 and not 35?? Please clarify this point.

On the top of page 5: Please indicate the unit of the median maximal dimension of malignant thyroid nodules (10 (7-13)).

Table 1: please reformate the points

Table 1: There are again 28 solitary and 8 multiple modules (please see my previous comment)

Table 2: please reformate the points

Table 4: please replace “ACR TI-RADS” by “ACR-TIRADS”

Table 5: Patient ID 3 is indicated with 18 years of age, but in your Materials and Methods section you wrote that the median age of your cohort was 15 (11-17)? Please clarify this point.

Discussion

This section is again well written and clearly to the point. The results are well discussed and the limitations of the presented study are indicated self-critically.

References

Covers all relevant current literature.

Grammar and style

The English used in the manuscript is adequate but would in some formulations benefit when proofread by a medical writer, an English editing service or a native speaker.

Thank you for your valuable contribution.

Author Response

b

Reviewer: 5

Comments to the Authors:

  1. With great interest I read the manuscript “Exploring the performance of ultrasound risk stratification systems in thyroid nodules of pediatric patients” submitted to Cancers. The study retrospectively evaluates the diagnostic performance of the main risk stratification systems for detecting malignant thyroid nodules in a study cohort of 36 under age patients with 41 thyroid nodules.The submitted manuscript is well designed and its methodology and statistical analysis are appropriate for its stated aim. The intention for the study is presented adequately and the results are clear-cut. It is well known, that pediatric thyroid nodules need a high clinical expertise for their higher malignancy risk compared to adults. Unfortunately, there are only few studies in current international literature evaluating diagnostic performance of the main risk stratification systems for detecting malignant thyroid nodules in children. The results of the submitted manuscript are interesting especially for endocrinologists, surgeons, radiologists and nuclear medicine physicians and therefore certainly worth publication. The results may also be a valuable help for the development of upcoming guidelines. With the presented study, the authors did high quality research. However, I have minor revisions concerning the manuscript.

Introduction

The introduction is clearly to the point and the problem is described adequately.

Materials & Methods

This section is well structured and covers all essential information.

Results

I don’t know if I got the following point right: You included 36 patients with 41 thyroid nodules (what is consistent in the manuscript). In the results section you wrote that “Among the 36 patients, 28 had a solitary thyroid nodule and eight had multiple thyroid nodules”. So if “multiple nodules” means more than 1 nodule there should be at least 16 nodules in these 8 patients. Added to the 28 solitary nodules this would be a total of 44 nodules? Please clarify this point.

Author: we greatly thank the reviewer for appreciating our work. It is correct that “Among the 36 patients, 28 had a solitary thyroid nodule and eight had multiple thyroid nodules” since not all the nodules of patients with 2 or more nodules were subjected to FNA.

  1. In figure 1 the text is not complete (at least in my pdf version). The upper box ends with “University of …??”, the lower box ends with “…at histopathology as ..:??” and the right box listing the excluded patients ends with “..nodules with inadequate…??”.

Authors: we greatly thank the reviewer for this remark. We do not know how it happened, since figure 1 was correctly uploaded. Maybe the automatic conversion by the journal from Arial to TimesNewRoman made these errors. Now, we directly corrected the figure 1 in the main text sent us by Cancers.

  1. Moreover, there are 35 patients excluded in figure 1: 14 with family history of thyroid cancer, 12 with indeterminate cytology without histology, 4 with previous malignancy, 3 with cancer genetic susceptibility syndroms, 3 with a history of radiation exposure and 2 nodules with inadequate??. 14+12+4+3+3+2 is 38 and not 35?? Please clarify this point.

Authors: we greatly thank the reviewer for this remark. The Figure 1 is correct since 14 patients with family history…..; 12 nodules wit indeterminate….; 4 patients with previous malignancy; 3 patients with cancer genetic….; 3 patients with a history…..; 2 nodules with inadequate…; 2 patients with lymph nodes….

  1. On the top of page 5: Please indicate the unit of the median maximal dimension of malignant thyroid nodules (10 (7-13)).

Authors: we greatly thank the reviewer for this suggestion. We added “mm".

  1. Table 1: please reformate the points

Authors: we thank the reviewer for this suggestion, but it is not possible to reformate the points in table 1 in the main text formatted by Cancers. Original Table 1 is formally correct and does not have irregular position of the points.

  1. Table 1: There are again 28 solitary and 8 multiple modules (please see my previous comment)

Authors: we thank the reviewer for this suggestion. As stated in item 3 Table 1 is correct regarding the number of patients with solitary and multiple nodules.

  1. Table 2: please reformate the points

Authors: we thank the reviewer for this suggestion, but, as for Table 1,also for Table 2 it is not possible to reformate the points in table 2 in the main text formatted by Cancers. Original Table 2 is formally correct and does not have irregular position of the points.

  1. Table 4: please replace “ACR TI-RADS” by “ACR-TIRADS”

Authors: we thank the reviewer for this remark. In Table 4“ACR TI-RADS” was replaced by “ACR-TIRADS.

  1. Table 5: Patient ID 3 is indicated with 18 years of age, but in your Materials and Methods section you wrote that the median age of your cohort was 15 (11-17)? Please clarify this point.

Authors: we thank the reviewer for this remark. Yes, this is correct since, as stated in statistical analysis, continuous variables were described as median and interquartile range (IQR).

  1. This section is again well written and clearly to the point. The results are well discussed and the limitations of the presented study are indicated self-critically. References covers all relevant current literature. Grammar and style. The English used in the manuscript is adequate but would in some formulations benefit when proofread by a medical writer, an English editing service or a native speaker. Thank you for your valuable contribution.

Authors: we greatly thank the reviewer for appreciating our work. A native English speaker validated our manuscript.

Reviewer: 5

Comments to the Authors:

  1. With great interest I read the manuscript “Exploring the performance of ultrasound risk stratification systems in thyroid nodules of pediatric patients” submitted to Cancers. The study retrospectively evaluates the diagnostic performance of the main risk stratification systems for detecting malignant thyroid nodules in a study cohort of 36 under age patients with 41 thyroid nodules.The submitted manuscript is well designed and its methodology and statistical analysis are appropriate for its stated aim. The intention for the study is presented adequately and the results are clear-cut. It is well known, that pediatric thyroid nodules need a high clinical expertise for their higher malignancy risk compared to adults. Unfortunately, there are only few studies in current international literature evaluating diagnostic performance of the main risk stratification systems for detecting malignant thyroid nodules in children. The results of the submitted manuscript are interesting especially for endocrinologists, surgeons, radiologists and nuclear medicine physicians and therefore certainly worth publication. The results may also be a valuable help for the development of upcoming guidelines. With the presented study, the authors did high quality research. However, I have minor revisions concerning the manuscript.

Introduction

The introduction is clearly to the point and the problem is described adequately.

Materials & Methods

This section is well structured and covers all essential information.

Results

I don’t know if I got the following point right: You included 36 patients with 41 thyroid nodules (what is consistent in the manuscript). In the results section you wrote that “Among the 36 patients, 28 had a solitary thyroid nodule and eight had multiple thyroid nodules”. So if “multiple nodules” means more than 1 nodule there should be at least 16 nodules in these 8 patients. Added to the 28 solitary nodules this would be a total of 44 nodules? Please clarify this point.

Author: we greatly thank the reviewer for appreciating our work. It is correct that “Among the 36 patients, 28 had a solitary thyroid nodule and eight had multiple thyroid nodules” since not all the nodules of patients with 2 or more nodules were subjected to FNA.

  1. In figure 1 the text is not complete (at least in my pdf version). The upper box ends with “University of …??”, the lower box ends with “…at histopathology as ..:??” and the right box listing the excluded patients ends with “..nodules with inadequate…??”.

Authors: we greatly thank the reviewer for this remark. We do not know how it happened, since figure 1 was correctly uploaded. Maybe the automatic conversion by the journal from Arial to TimesNewRoman made these errors. Now, we directly corrected the figure 1 in the main text sent us by Cancers.

  1. Moreover, there are 35 patients excluded in figure 1: 14 with family history of thyroid cancer, 12 with indeterminate cytology without histology, 4 with previous malignancy, 3 with cancer genetic susceptibility syndroms, 3 with a history of radiation exposure and 2 nodules with inadequate??. 14+12+4+3+3+2 is 38 and not 35?? Please clarify this point.

Authors: we greatly thank the reviewer for this remark. The Figure 1 is correct since 14 patients with family history…..; 12 nodules wit indeterminate….; 4 patients with previous malignancy; 3 patients with cancer genetic….; 3 patients with a history…..; 2 nodules with inadequate…; 2 patients with lymph nodes….

  1. On the top of page 5: Please indicate the unit of the median maximal dimension of malignant thyroid nodules (10 (7-13)).

Authors: we greatly thank the reviewer for this suggestion. We added “mm".

  1. Table 1: please reformate the points

Authors: we thank the reviewer for this suggestion, but it is not possible to reformate the points in table 1 in the main text formatted by Cancers. Original Table 1 is formally correct and does not have irregular position of the points.

  1. Table 1: There are again 28 solitary and 8 multiple modules (please see my previous comment)

Authors: we thank the reviewer for this suggestion. As stated in item 3 Table 1 is correct regarding the number of patients with solitary and multiple nodules.

  1. Table 2: please reformate the points

Authors: we thank the reviewer for this suggestion, but, as for Table 1,also for Table 2 it is not possible to reformate the points in table 2 in the main text formatted by Cancers. Original Table 2 is formally correct and does not have irregular position of the points.

  1. Table 4: please replace “ACR TI-RADS” by “ACR-TIRADS”

Authors: we thank the reviewer for this remark. In Table 4“ACR TI-RADS” was replaced by “ACR-TIRADS.

  1. Table 5: Patient ID 3 is indicated with 18 years of age, but in your Materials and Methods section you wrote that the median age of your cohort was 15 (11-17)? Please clarify this point.

Authors: we thank the reviewer for this remark. Yes, this is correct since, as stated in statistical analysis, continuous variables were described as median and interquartile range (IQR).

  1. This section is again well written and clearly to the point. The results are well discussed and the limitations of the presented study are indicated self-critically. References covers all relevant current literature. Grammar and style. The English used in the manuscript is adequate but would in some formulations benefit when proofread by a medical writer, an English editing service or a native speaker. Thank you for your valuable contribution.

Authors: we greatly thank the reviewer for appreciating our work. A native English speaker validated our manuscript.

Round 2

Reviewer 3 Report

In my view, the manuscript is unacceptable as it stands, because this research has serious flaws - as in the previous version.